# Thermo-Acoustic Catalytic Effect on Oxidizing Woody Torrefaction

**Edgar A. Silveira [1],\*, Luiz Gustavo Oliveira Galvão [2], Lucélia Alves de Macedo [2], Isabella A. Sá [3], Bruno S. Chaves [2], Marcus Vinícius Girão de Morais [1] , Patrick Rousset [4] and Armando Caldeira-Pires [1]**

[1] Mechanical Engineering Department, University of Brasília, Brasilia DF 70910-900, Brazil; mvmorais@unb.br (M.V.G.d.M.); armandcp@unb.br (A.C.-P.)

[2] Forest Products Laboratory, Brazilian Forest Service, Brasilia 70818900, Brazil; luiz.galvao@florestal.gov.br (L.G.O.G.); lucelia.macedo@florestal.gov.br (L.A.d.M.); bruno.chaves@florestal.gov.br (B.S.C.)

[3] Forest Engineering Department, University of Brasília, Brasilia DF 70910-900, Brazil; isabellaasa29@gmail.com

[4] BioWooEB, French Agriculture Research Centre for International Development (CIRAD), 73 rue J. F. Breton, 34398 Montpellier, CEDEX 5, France; patrick.rousset@cirad.fr

\* Correspondence: edgar.silveira@unb.br

**Abstract:** The torrefaction (mild pyrolysis) process modifies biomass chemical and physical properties and is applied as a thermochemical route to upgrade solid fuel. In this work, the catalytic effect of thermo-acoustic on oxidizing woody torrefaction is assessed. The combined effect of two acoustic frequencies (1411, 2696 Hz) and three temperatures (230, 250, and 290 °C) was evaluated through weight loss and its deviation curves, calculated torrefaction severity index (TSI), as well as proximate, calorific, and compression strength analysis of *Eucalyptus grandis*. A new index to account for the catalytic effects on torrefaction (TCEI) was introduced, providing the quantitative analysis of acoustic frequencies influence. A two-step consecutive reaction numerical model allowed the thermo-acoustic experiment evaluation. For instance, the thermogravimetric profiles revealed that the acoustic field has a catalytic effect on wood torrefaction and enhances the biomass oxidation process for severe treatments. The kinetic simulation of the acoustic coupling resulted in faster conversion rates for the solid pseudo-components showing the boosting effect of acoustic frequencies in anticipating hemicellulose decomposition and enhancing second step oxidizing reaction.

**Keywords:** woody biomass torrefaction; thermoacoustic; catalytic effect; numerical modeling; severity factors

## 1. Introduction

Renewable energy accounts for 45.2% of the Brazilian internal energy supply [1]. Biomass (firewood and charcoal) is the third most representative renewable source (8.4%) [1]. Biomass is considered to be a carbon-neutral fuel from a climate change perspective [2]. Nevertheless, biomass as a raw material has less energy density, more moisture, and volatiles than coal [3]. Energetical conversion process (pyrolysis, torrefaction, and gasification) are required to overcome these issues and upgrade biofuels replacing more conventional energy sources [4]. Torrefaction appears as a promising thermochemical conversion route. In this process, the biomass is heated at 200–300 °C in a partial or total absence of oxygen, aiming to produce a solid fuel more homogeneous, hydrophobic, and with a higher carbon content than the raw material [3,4].

Major factors governing chemical reactions in a porous medium include chemical kinetics and heat and mass transfer, which are decisive in many devices of the chemical and energy industries [5]. Given the development of thicker boundary layers by larger-sized biomass particles during torrefaction, some limitations arise from problems related to heat and mass transfer issues [6]. The motivation for this research work came from the potential use of this wood thermal treatment combined with an acoustic system to improve torrefaction treatment [5].

Some studies on heat and mass transfer showed connections between acoustic energy and thermal processes. The phenomena as combustion instabilities [7,8], Rijke tubes, [9] and thermoacoustic heat engines [10,11] are thermoacoustic problems well explored. The attempt to enhance heat transfer by modifying the interaction between solids and ambient gas by applying a strong acoustic field was discussed in [12,13]. Results showed that the heat transfer rate between a preheated wire and ambient gas could be enhanced under the application of sound waves. The heat transfer coefficient increases with the sound strength in both standing and traveling sound waves.

New technologies coupled to thermal modification torrefaction reactors as a vacuum atmosphere [14–17], microwaves [18–21], wet-torrefaction [22,23], and potassium impregnation [24–26] have been explored to improve the thermal pre-treatment. Some studies with ultrasound for biomass pre-treatment explore the sonochemical and mechanoacoustic pre-treatments effects [23]. The mechanoacoustic effect alters the surface structure of the biomass, while the sonochemical production of oxidizing radicals leads to a chemical attack of the components [23]. Yang et al., (2019) explored two fuel upgrading actions (torrefaction and pelleting) simultaneously with the assistance of ultrasonic vibration [27]. It was found that the torrefied pellets had improved physical, thermochemical, and hygroscopic properties and increased carbon content, indicating higher heating values (HHV) of torrefied pellets over non-torrefied biomass [27].

Our earlier study proposed a thermo-acoustic torrefaction reactor [5]. This study showed that temperature coupled to acoustic frequencies modified the interaction between the oxidative atmosphere and wood surface and a catalytic behavior for solid yield evolution and higher conversion rates was reported [5]. Based on these findings, this work aimed to investigate the acoustic catalytic effect on the torrefaction treatment dynamics and the torrefied biomass properties [5], providing a better understanding of the innovative technological process.

## 2. Material and Methods

### 2.1. Feedstock and Sample Analysis

In this work, a common hardwood species used in Brazil, *Eucalyptus grandis,* was employed due to its industrial importance. A 15-year old monitored growing tree was harvested and cut into $3 \times 3 \times 3$ cm blocks with approximately the same physical characteristics (density, size, and mass) [5]. Before the experiments, the cubic samples were arranged in an oven at 105 °C until dry conditions were obtained [5]. The anhydrous weight ($w_0$) was determined at the end of the drying process.

The proximate, calorific, and mechanical compression analyses were conducted on raw and torrefied samples to examine its basic properties and how the thermoacoustic affects the final product. All analyses were made in duplicate. The fixed carbon (FC), volatile matter (VM), and ash contents were determined in grinded samples sieved to a particle size of <60 mesh, following the ISO standard 18123:2015 and ISO 18122 procedure. Raw and torrefied biomass HHVs were obtained with a PARR 6400 bomb calorimeter, according to ISO standard ISO 18125.

Compression strength (CS) analysis allowed an indirect assessment of the wood friability under different temperatures and thermoacoustic treatments. The CS was determined with a Universal Testing Machine, Model DL30000, from EMIC. The cubic samples were compressed at a continuous rate of 0.6 mm·min$^{-1}$ until failure. The maximum load of the wood cube rupture was calculated following the ASTM D143/2000 standard. The proximate, calorific, and compression strength results for the raw material are shown in Table 1.

**Table 1.** Proximate, calorific, and compression strength analyses of *Eucalyptus grandis*.

| Raw Material | *Eucalyptus grandis* |
|---|---|
| Proximate analysis [a] | |
| Fixed carbon (%) | 18.51 |
| Volatile matter (%) | 81.4 |
| Ash (%) | 0.09 |
| HHV (MJ kg$^{-1}$) | 20.09 |
| Max. load (kgf) | 10,874.16 |

[a] Dry basis.

## 2.2. Thermal Gravimetric (TG) Experimental Analysis

The thermal gravimetric (TG) behavior during the torrefaction experiment was evaluated through a series of experiments carried out in a developed thermoacoustic reactor [5]. Reactor setup and acoustics details are described in a previous study [5]. The acoustic system, coupled with the existing reactor [10], is illustrated in Figure 1.

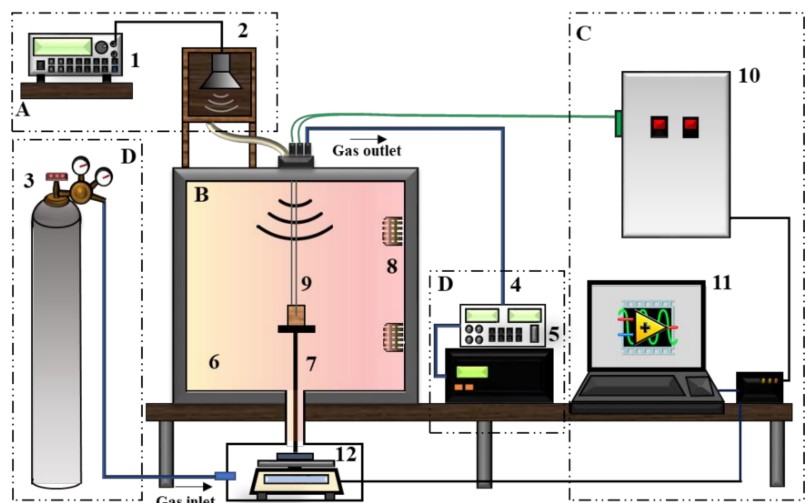

**Figure 1.** General diagram of the thermoacoustic torrefaction system. The device involved four subsystems: acoustic (A), heat treatment (B), power and recording (C), and gas feeding (D). System components: (**1**) Wave generator; (**2**) Sound speaker; (**3**) N2 cylinder; (**4**) Gas pump; (**5**) $O_2$ control; (**6**) Reactor chamber; (**7**) Wood sample support; (**8**) Electric resistances; (**9**) Thermocouples; (**10**) System control; (**11**) Computer; and (**12**) Electric weight balance [5].

The acoustic subsystem (A) was comprised of a wave generator and one speaker connected by a flexible duct to transmit the acoustic energy to the reactor [5]. The oxidative atmosphere was controlled by $N_2$ injection. The heat treatment subsystem (B) was composed of a reactor chamber, a support for the wood sample connected to a mass balance (Sartorius LP2200S) with an accuracy of $10^{-3}$ grams, two electric resistances and two type K special thermocouples (*IEC 584-3*) with a bead size of 1 mm and a tolerance value of 1.1 °C to determine the temperatures of the wood surface and wood core [5]. To control the reactor temperatures and heating rate, a programmable proportional-integral-derivative (PID) was utilized based on data from a thermal sensor PT100 placed within the system in the center of the reactor to record the atmosphere temperature [5]. The system provides continued acquisition data with a 100 Hz sampling rate (e.bloxx A4-1TC Multichannel) recording thermocouples temperature profiles and mass weight during the wood heat treatment [5].

Torrefaction treatment was conducted for 60 min and evaluated for the three severity classifications, Light, Mild, and Severe according to a previous study [2]. The cubic samples were heated at a linear heating rate of 5 °C min$^{-1}$ until the temperatures of 230, 250, and 290 °C were obtained. The oxidizing

atmosphere of 10% $O_2$ was chosen due to increased operating costs with inert gas supply [28] and continuously controlled by $N_2$ (99.99%) injection within the reactor chamber [5]. A triplicate was performed for each combination of temperature and frequency.

The control experiments were performed without acoustic conditions for all temperatures. The thermoacoustic torrefaction sets of experiments were performed for the three temperatures coupled to the identified 1411, 2696 Hz acoustic frequencies in a previous study [5]. Within the system limits, those frequencies can produce the maximum superficial velocity around the wood sample affecting the interaction between the oxidizing gaseous environment and wood surface [5]. Each frequency was sustained throughout the full experimentation [5]. The temperature and frequency coupling effect was assessed by the TGA, its deviation (DTG) curves, and torrefied solid product analysis. The torrefaction treatment parameters are listed in Table 2.

**Table 2.** Thermo-acoustic torrefaction parameters.

| Thermoacoustic Torrefaction Conditions | |
| --- | --- |
| **Raw Material** | *Eucalyptus grandis* |
| Duration | 60 min |
| Heating rate | 5 °C min$^{-1}$ |
| Atmosphere | 10% $O_2$ |
| Final temperature/acoustic frequency | |
| 230 °C [a]/- | |
| 230 °C/1411 Hz | |
| 230 °C/2696 Hz | |
| 250 °C [a]/- | |
| 250 °C/1411 Hz | |
| 250 °C/2696 Hz | |
| 290 °C [a]/- | |
| 290 °C/1411 Hz | |
| 290 °C/2696 Hz | |

[a] Control experiment.

The normalized weight loss evolution over time for each temperature is expressed as $TG_{T,exp}^{(F)}(t)$ and was obtained by the calculated ratio between the continuously weighted wood sample $w_i(t)$ and anhydrous weight ($w_0$), according to Equation (1) [29].

$$TG_{T,exp}^{(F)}(t) = \frac{w_i(t)}{w_0} \times 100 \tag{1}$$

where $T$ is the treatment temperature (230, 250, and 290 °C); $F$ is the acoustic condition of control (no acoustic), 1411, and 2696 Hz; and $t$ is the continuous treatment time.

*2.3. Thermoacoustic Kinetics Modeling*

The two-step, first-order mechanism kinetics model, originally proposed [30] and modified for thermal sensitivity analysis [29,31], was conducted in this study. Modeling inputs and boundary conditions are detailed in past studies [29,31,32]. The two consecutive step reactions of Di Blasi and Lanzetta (1997) model are represented by three solid pseudo-components $A$ (feedstock), $B$ (intermediate solid), and $C$ (residue) and two volatiles $V_1$ and $V_2$ as Equations (2) and (3) [29]:

$$\text{1st reaction step} : \begin{cases} A \xrightarrow{k_1} B \\ A \xrightarrow{k_{V_1}} V_1 \end{cases} \tag{2}$$

$$\text{2nd reaction step}: \begin{cases} B \xrightarrow{k_2} C \\ B \xrightarrow{k_{V_2}} V_2 \end{cases} \tag{3}$$

The modeling provided the numerical calculation of the kinetic rates ( $k_1$, $k_2$, $k_{V_1}$, $k_{V_2}$) to analyze the influence of temperature and acoustic frequency into thermoacoustic torrefaction. The numerical calculate solid yield $SY_{T,cal}^{(F)}(t)$ is obtained by the sum of $Y_{T,A}^{(F)}(t)$, $Y_{T,B}^{(F)}(t)$ and $Y_{T,C}^{(F)}(t)$ [29,31–33] using Equations (4)–(7).

$$Y_{T,A}^{(F)}(t) = \frac{dm_A(t)}{dt} = -\left(k_1 + k_{V_1}\right) \times m_A(t) \tag{4}$$

$$Y_{T,B}^{(F)}(t) = \frac{dm_B(t)}{dt} = k_1 \times m_A(t) - \left(k_2 + k_{V_2}\right) \times m_B(t) \tag{5}$$

$$Y_{T,C}^{(F)}(t) = \frac{dm_C(t)}{dt} = k_2 \times m_B(t) \tag{6}$$

$$SY_{T,cal}^{(F)}(t) = Y_{T,A}^{(F)}(t) + Y_{T,B}^{(F)}(t) + Y_{T,C}^{(F)}(t) \tag{7}$$

The released volatiles are described in Equations (8) and (9) by the sum of $V_1$ and $V_2$ [31–33].

$$Y_{T,V_1}^{(F)}(t) = \frac{dm_{V1}(t)}{dt} = k_{V_1} \times m_A(t) \tag{8}$$

$$Y_{T,V_2}^{(F)}(t) = \frac{dm_{V2}(t)}{dt} = k_{V_2} \times m_B(t) \tag{9}$$

The Arrhenius kinetic parameters ( $k_1$, $k_2$, $k_{V1}$, $k_{V2}$) provided quantitative information about the reaction rates. These parameters are determined with a minimization function in a Matlab® routine by fitting the numerical calculate solid yield $SY_{cal}^{(T)}(t)$ to experimental TG curves evolution $TG_{T,exp}^{(F)}(t)$ according to Equation (10) [29]:

$$k_i = k_{0,i} exp\left(\frac{-E_{a,i}}{RT}\right) \tag{10}$$

where $E_{a,i}$ (J·mol$^{-1}$) and $k_{0,i}$ (min$^{-1}$) are respectively the activation energies and the pre-exponential factors of the reactions, $R$ is the universal gas constant (J·mol$^{-1}$·K$^{-1}$), and $T$ is the absolute temperature (K) [29].

## 3. Results and Discussions

The thermo-acoustic torrefaction results are presented in Section 3.1 for the 230, 250, and 290 °C temperatures. In Section 3.2, the effect of thermoacoustic treatment on physical and chemical properties is exposed. Section 3.3 describes the obtained result for the kinetic simulation analysis.

*3.1. Thermoacoustic Experiments*

3.1.1. TG and DTG Curve Assessment

For the acoustic catalytic effect analysis on the thermal process, the TG and DTG profiles' evolution over time under the influence of 1411 and 2696 Hz frequencies are shown in Figure 2a,d, Figure 2b,e, and Figure 2c,f for 230, 250, and 290 °C, respectively. Previous studies showed that lignocellulosic biomass degradation takes place between 180–200 °C [34]. For better interpretation, the $TG_{T,exp}^{(F)}(t)$ profiles were normalized at 170 °C [29].

Torrefaction severity highly influences wood component degradation [33]. The thermal stability for control experiments (without acoustic) agrees with the literature [35,36], starting degradations at 181.7 °C. The control TG profile becomes rough with higher temperatures and the final product solid yield is 93.9, 87.2, 66.2% for Light (230 °C), Mild (250 °C), and Severe (290 °C), respectively, agreeing with previous studies [14,18].

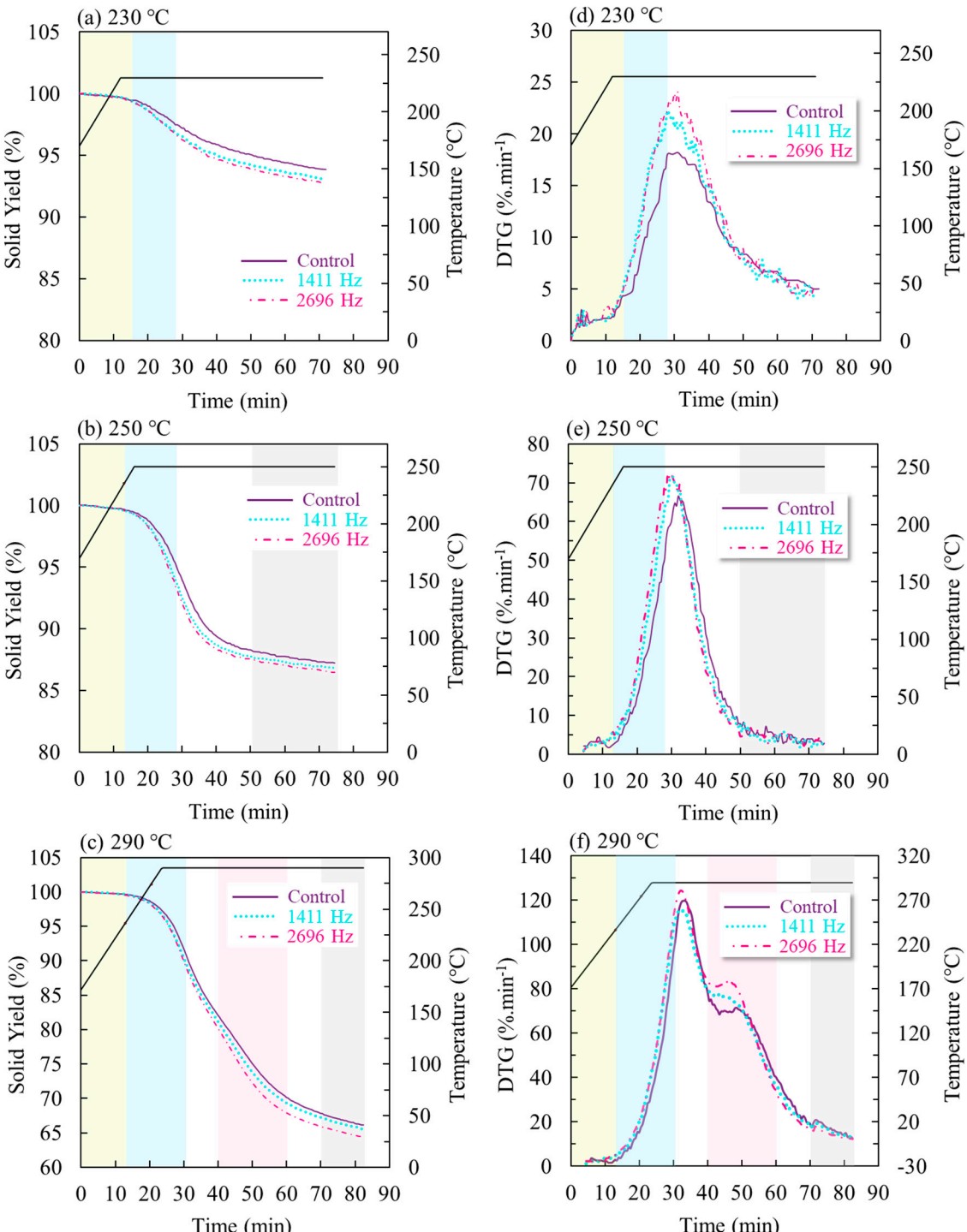

**Figure 2.** Thermogravimetric (TG) and deviation of thermo-gravimetric (DTG) profiles of thermoacoustic torrefaction for control, 1411, and 2696 Hz under (**a**,**d**) 230, (**b**,**e**) 250, and (**c**,**f**) 290 °C temperature treatments.

For both acoustic frequencies, an earlier degradation was evidenced. Regarding 230, 250, and 290 °C TG curves, the catalytic effect starts at around 14.5 min for both acoustic frequencies. However, the resulting values for the final solid yield did not show significant differences, agreeing with the first results [5].

Analyzing the DTG curves, two peaks were identified. The 1st DTG denotes the conversion rate of the solid wood into volatile compounds. The 1st DTG peaks were found in different treatment times

between 20 and 40 min. It was observed that the catalytic effect of acoustic in enhancing degradation rates is more evident at 230 °C, decreasing with increasing temperatures. Analyzing 230 °C torrefaction, 1st DTG peaks intensities were 18.17 (31 min), 22.26 (28 min), 24.07 (30 min) %·min$^{-1}$, respectively for control, 1411, and 2696 Hz. For 250 °C, intensities were 66.62 (32 min), 72.52 (29 min), and 71.49 (30 min) %·min$^{-1}$ for control, 1411, and 2696 Hz, respectively.

Regarding 290 °C, the effect of acoustic was not observed at the 1st peak (at about 35 min), with intensities around 120% min$^{-1}$ for control, 1411, and 2696 Hz. However, as the treatment time progressed, samples displayed a slight 2nd DTG peak at 46 min. The catalytic acoustic effect can be observed at this 2nd DTG peak for both thermoacoustic experiments, becoming greater for higher frequencies (Figure 2f). The 2nd DTG peaks exhibit intensities of 69.56, 75.80, and 83.97%·min$^{-1}$ for control, 1411, and 2696 Hz, respectively. This second stage of decomposition could be related to biomass oxidation, occurring successively to ordinary torrefaction [37]. Therefore, for the conditions tested in this work (10% $O_2$ and cubic particle size of 3 cm), 290 °C seems to be the temperature where the biomass oxidation becomes significant.

According to a previous work [38], this oxidation stage during oxidative torrefaction is dominated by surface oxidation, which intensifies the internal heat and mass transfer when the temperature and superficial velocity rise. Higher temperatures within the wood sample (core) were evidenced for higher temperature treatments during thermoacoustic torrefaction [5]. Therefore, the catalytic effect of acoustic in increasing oxidation could be due to faster particle velocities around the wood sample, modifying the interaction between released volatiles and the sample's surface, promoting an increase in the superficial area, thus intensifying internal heat and mass transfer because of the higher surface oxidation.

Even if further studies are still needed to determine the real potential applicability of acoustic frequencies coupled to torrefaction, this approach holds the promise of conducting thermal treatment within shorter durations or lower temperatures, especially under oxidative atmospheres, reducing, therefore, the energy consumption in torrefaction plants.

### 3.1.2. Torrefaction Severity Index (TSI) for the Catalytic Process

The TSI accounts for the severity degree of the biomass at different torrefaction conditions [39]. The torrefaction severity index, firstly proposed in a study [39], was shown to integrate all TG physical meaning, being suitable to perform different torrefaction condition comparisons and mitigating the weight loss variability. For this study, the TSI was analyzed for each temperature separately. The TSI was calculated for the three acoustic conditions, no acoustic (control), 1411, and 2696 Hz. The TG experiments (Figure 2) showed that the catalytic effect was higher for 2696 Hz, this acoustic condition being the reference as the tough treatment at the final treatment time $t_f$. The TSI was defined for each temperature as Equation (11). The TSI displays the torrefaction severity normalized between 0 and 1 for the three temperatures, allowing a better comparison between Light, Mild, and Severe torrefaction. To obtain a qualitative catalytic behavior pattern between thermo-acoustic treatments, the adimensional torrefaction catalytic effect index (TCEI) was introduced and defined as Equation (12):

$$TSI_T^F(t) = \frac{100 - TG_{T,exp}^{(F)}(t)}{100 - TG_{T,exp}^{(2696)}(t_f)} \tag{11}$$

$$TCEI_T^F(t) = 1 - \frac{TSI_T^F(t)}{TSI_T^{control}(t)} \tag{12}$$

In Figure 3, the calculated TSI and TCEI indexes are displayed, allowing a quantitative assessment of the catalytic acoustic effect.

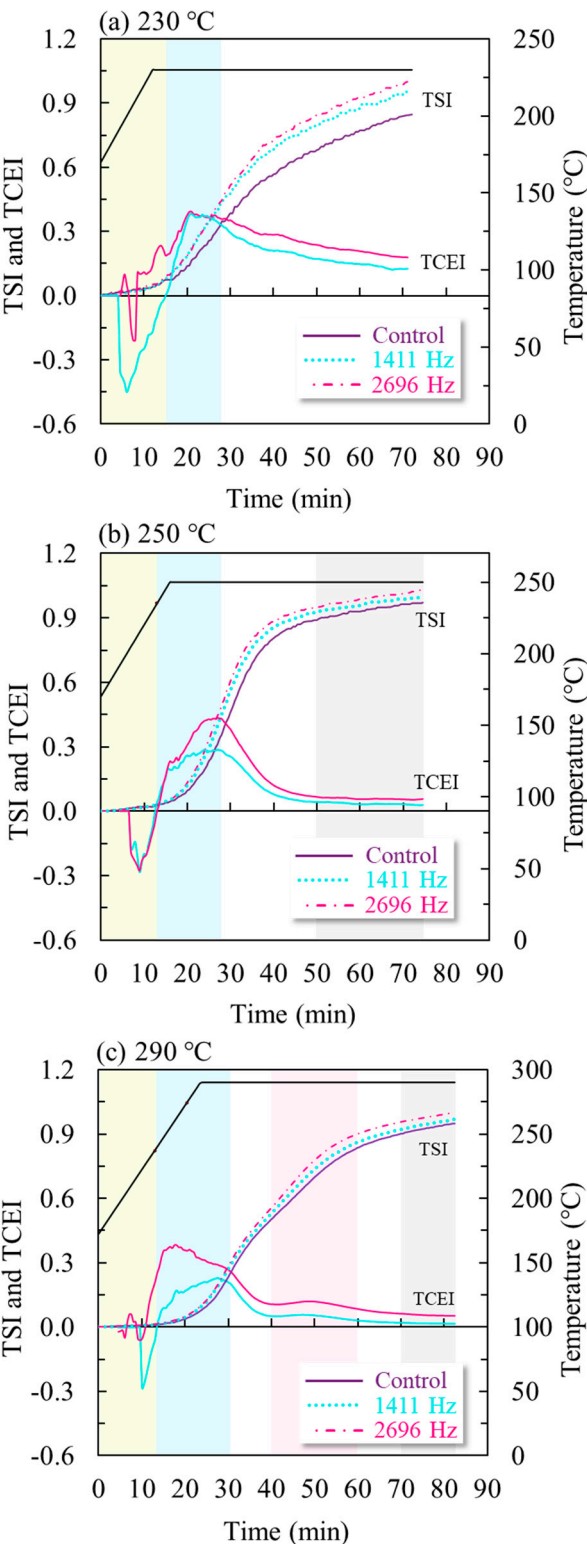

**Figure 3.** Torrefaction severity index (TSI) and torrefaction catalytic effect index (TCEI) profiles of thermoacoustic torrefaction for control, 1411, and 2696 Hz under (**a**) 230, (**b**) 250, and (**c**) 290 °C temperature treatments.

The TSI curvatures for acoustic treatment presented greater slopes compared to control. Physically, the higher the curvature in the profile, the more sensitive the biomass weight loss to torrefaction operation is [39]. Four distinguishing regions were identified with the index analysis and are presented

in Figures 2 and 3. The green region comprehends the first 14.5 min and temperatures between 170–235 °C, where degradations start, and weight loss promotes dissimilar oscillations. The blue region emphasizes the first acoustic catalytic zone, having different durations: 14.5–27 min for light and mild and 14.5–31 min for severe torrefaction. Within the blue zone in Figure 3, the TCEI values increase, reach a maximum, followed by a decrease. The first catalytic zone is evidenced in Figure 2 by the earlier solid degradation and the anticipated and greater 1st DTG peaks (hemicelluloses degradation).

A third region (pink) between 40–60 min is exclusive of severe torrefaction (290 °C), presenting the second catalytic effect region. Regarding the 290 °C thermoacoustic experiments, a rough TSI profile slope and a minor second peak of TCEI are evidenced in Figure 3c. In Figure 2, the pink region depicts the 2nd DTG peak (enhancing cellulose oxidation) and the increase between weight loss profile distances.

For instance, TCEI maximum values (blue region) for light torrefaction were 0.3828 and 0.3947, for 1411 and 2696 Hz, respectively. For 250 and 290 °C temperatures, the maximum values were 0.287, 0.230 for 1411 Hz, and 0.437, 0.381 for 2696 Hz. The 250 °C temperature showed higher values compared to 290 °C temperature within the first catalytic effect blue zone. Nevertheless, the 290 °C temperature had extended catalytic effect influence during treatment. The gray region in the treatment's end is characterized by TCEI flat profiles, showing no enhancing on catalytic effect and no changes between the solid yield.

### 3.2. Physical and Chemical Property Analysis

The obtained values for the volatile matter (VM), fixed carbon (FC), and HHV are presented in Figure 4a–c, respectively. The higher the temperature treatments, the lower the volatiles values and the greater fixed carbon, and consequently, HHV. The VM, FC, and HHV results for control treatment are according to reported literature values [28].

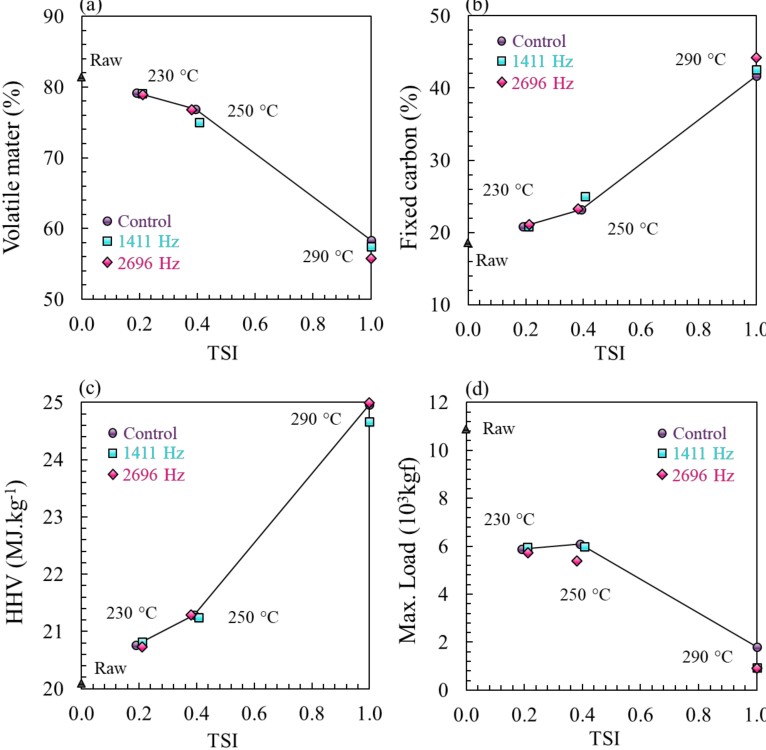

**Figure 4.** Results for (**a**) volatile mater, (**b**) fixed carbon, (**c**) higher heating values (HHV), and (**d**) compression strength in function of TSI for control, 1411, and 2696 Hz under 230, 250, and 290 °C temperature treatments.

As evaluated in Figure 2, the solid yield value does not vary expressively, being the catalytic effect of acoustic noticeable during the treatment, not at this end [5]. Therefore, the final product analysis results did not show significant differences. The compression strength (CS) was analyzed and is displayed in Figure 4d. The previous study reported a higher core temperature for acoustic treatment [5]. The 290 °C acoustics experiment results for CS were lower than control. This behavior might be attributed to higher thermal stress of the woody structure due to the higher temperature gradients. The catalytic effect could affect the anisotropic woody structure due to the fast conversion rates of wood components (hemicelluloses) and cellulose oxidation [37]. The thermoacoustic torrefaction conferred an improved torrefied product grindability. This leads to more efficient processes enhancing added final value [40].

*3.3. Thermoacoustic Kinetic Modeling*

The calculated solid yield $SY_{T,cal}^{(F)}(t)$ of the three acoustic conditions is displayed in Figure 5a–c for the 230, 250, and 290 °C torrefaction, respectively. The obtained pre-exponential factors and activation energies of kinetic parameters for thermoacoustic torrefaction are in Table 3, and the Arrhenius plot is displayed in Figure 5d.

**Table 3.** The obtained kinetic parameters under control and thermoacoustic torrefaction experiments (1411 and 2696 Hz) for 230, 250, and 290 °C.

| | Reaction Kinetic Constant $K_i$ | $A{\rightarrow}B$ $K_1$ | $A{\rightarrow}V_1$ $K_{V_1}$ | $B{\rightarrow}C$ $K_2$ | $B{\rightarrow}V_2$ $K_{V_2}$ |
|---|---|---|---|---|---|
| Control | $E_{a,i}$ (kJ·kg$^{-1}$) | 106,757 | 139,152 | 35,434 | 129,916 |
| | $k_{o,i}$ (min$^{-1}$) | $2.85 \times 10^9$ | $4.55 \times 10^{11}$ | 32.61 | $8.22 \times 10^9$ |
| | 230 °C | $2.35 \times 10^{-2}$ | $1.63 \times 10^{-3}$ | $6.84 \times 10^{-3}$ | $2.68 \times 10^{-4}$ |
| | 250 °C | $6.24 \times 10^{-2}$ | $5.81 \times 10^{-3}$ | $9.45 \times 10^{-3}$ | $8.78 \times 10^{-4}$ |
| | 290 °C | $3.57 \times 10^{-1}$ | $5.64 \times 10^{-2}$ | $1.69 \times 10^{-2}$ | $7.33 \times 10^{-3}$ |
| 1411 Hz | $E_{a,i}$ (kJ·kg$^{-1}$) | 102,176 | 152,936 | 42,891 | 142,169 |
| | $k_{o,i}$ (min$^{-1}$) | $1.66 \times 10^9$ | $1.87 \times 10^{13}$ | 229.56 | $1.37 \times 10^{11}$ |
| | 230 °C | $4.09 \times 10^{-2}$ | $2.48 \times 10^{-3}$ | $8.10 \times 10^{-3}$ | $2.39 \times 10^{-4}$ |
| | 250 °C | $1.04 \times 10^{-1}$ | $1.00 \times 10^{-2}$ | $1.20 \times 10^{-2}$ | $8.77 \times 10^{-4}$ |
| | 290 °C | $5.52 \times 10^{-1}$ | $1.22 \times 10^{-1}$ | $2.41 \times 10^{-2}$ | $8.94 \times 10^{-3}$ |
| 2696 Hz | $E_{a,i}$ (kJ·kg$^{-1}$) | 101,122 | 152,864 | 42,558 | 141,767 |
| | $k_{o,i}$ (min$^{-1}$) | $1.58 \times 10^9$ | $2.14 \times 10^{13}$ | 238.45 | $1.53 \times 10^{11}$ |
| | 230 °C | $5.03 \times 10^{-2}$ | $2.89 \times 10^{-3}$ | $9.11 \times 10^{-3}$ | $2.94 \times 10^{-4}$ |
| | 250 °C | $1.27 \times 10^{-1}$ | $1.17 \times 10^{-2}$ | $1.34 \times 10^{-2}$ | $1.07 \times 10^{-3}$ |
| | 290 °C | $6.60 \times 10^{-1}$ | $1.42 \times 10^{-1}$ | $2.69 \times 10^{-2}$ | $1.09 \times 10^{-2}$ |

$E_{a,i}$ : Activation energies; $k_{o,i}$ : pre-exponential factors ($i = 1, 2, V_1 \ and \ V_2$).

The thermoacoustic TG profiles over time served as input data for the Matlab® routine to predict solid yield and obtain reaction rate competition [29]. The resulting solid yield displays an accurate fitting with experimental data for treatment beginning and end, agreeing with previous studies [29,32]. The chosen three-stage approach can allow to obtain decent results for the whole torrefaction range for macroparticles of *Eucalyptus grandis,* as expected [29]. The predicted curves could reduce experimental procedures and provide valuable insights and information concerning treatment residence time, conversion rates, and biomass thermal degradation mechanisms for the torrefaction industry.

The thermoacoustic reaction rates were faster than the controls, the first step being more distinguished than the second step. The reaction rate ranking had two behaviors. For light torrefaction, the ranking was $k_1 > k_2 > k_{v1} > k_{v2}$. The 1st reaction step was faster than the second for Mild and Severe torrefaction showing a competition rank of $k_1 > k_{v1} > k_2 > k_{v2}$, as pointed out by previous results [29]. Figure 5d shows that the transition between both aforementioned competition rate behaviors was anticipated for acoustic treatments, taking place at 250 °C, while for control, it took place around 260 °C. Figure 5d shows that the first step is highly enhanced (hemicellulose) for acoustic conditions, with a lesser enhancement of the second step (cellulose oxidation).

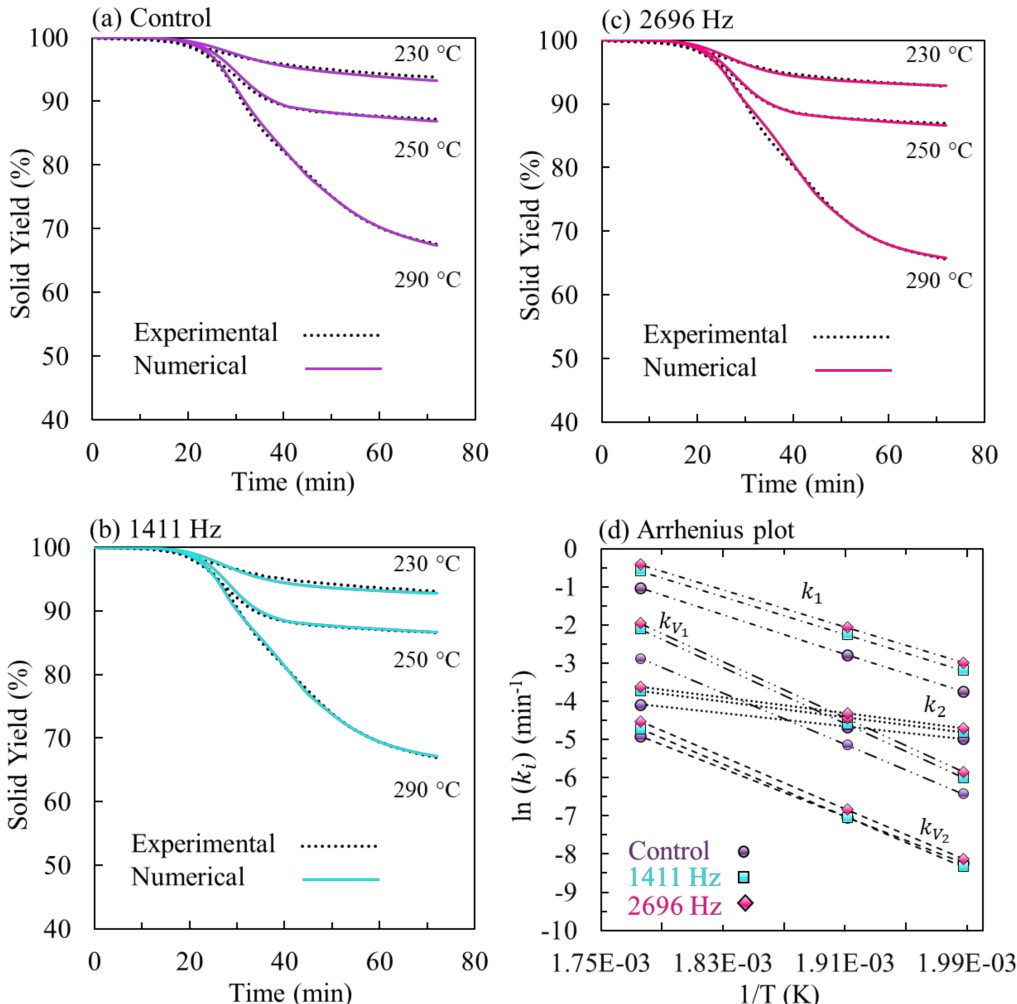

**Figure 5.** Simulated and experimental curve fitting for (**a**) control and acoustic treatments (**b**) 1411 and (**c**) 2696 Hz performed at 230, 250, and 290 °C temperature treatments. (**d**) Arrhenius plot of reaction rates for control, 1411, and 2696 Hz treatments.

The solid and volatiles pseudo-component evolution for 230, 250, and 290 °C treatments are displayed in Figure 6. The catalytic effect is clearly for the three conditions, the catalysis effect being ranked as 2696 Hz > 1411 Hz > control, agreeing with experimental results. The identified regions during an index analysis were displayed for a better interpretation of pseudo-components' physical meaning.

Regarding the 230 °C treatment, $A$ was not entirely consumed for the control experiment, remaining at 16.17%. Meanwhile, for the acoustic treatments, $A$ was almost wholly consumed and can be compared to the 250 °C control treatment in terms of degradation rate. The intermedia solid $B$ had 58.54% of the final solid product and, differently of both acoustic treatments, did not start its decreasing. The $C$ evolution started around 20 min with an almost linear increase, the final acoustic values being higher than the control ensuing frequency intensity. The catalytic effect (blue region) started after the green region (14.5 min) with $A$ decomposition, $B$ formation, and $V_1$ releasing, having acoustic treatment faster rates than control. For lower temperatures, the hemicelluloses were mainly consumed [33], having almost no $V_2$ volatile group and smaller values for $V_1$.

During 250 and 290 °C, a faster and earlier $B$ formation followed by its higher consumption was observed, leading to an earlier release of $V_1$, therefore a smaller extent for the acoustic treatments at the final step of the treatment compared to control. The $C$ and $V_2$ profiles showed rough slopes, the acoustic ones being faster and anticipated for Mild and Severe treatments.

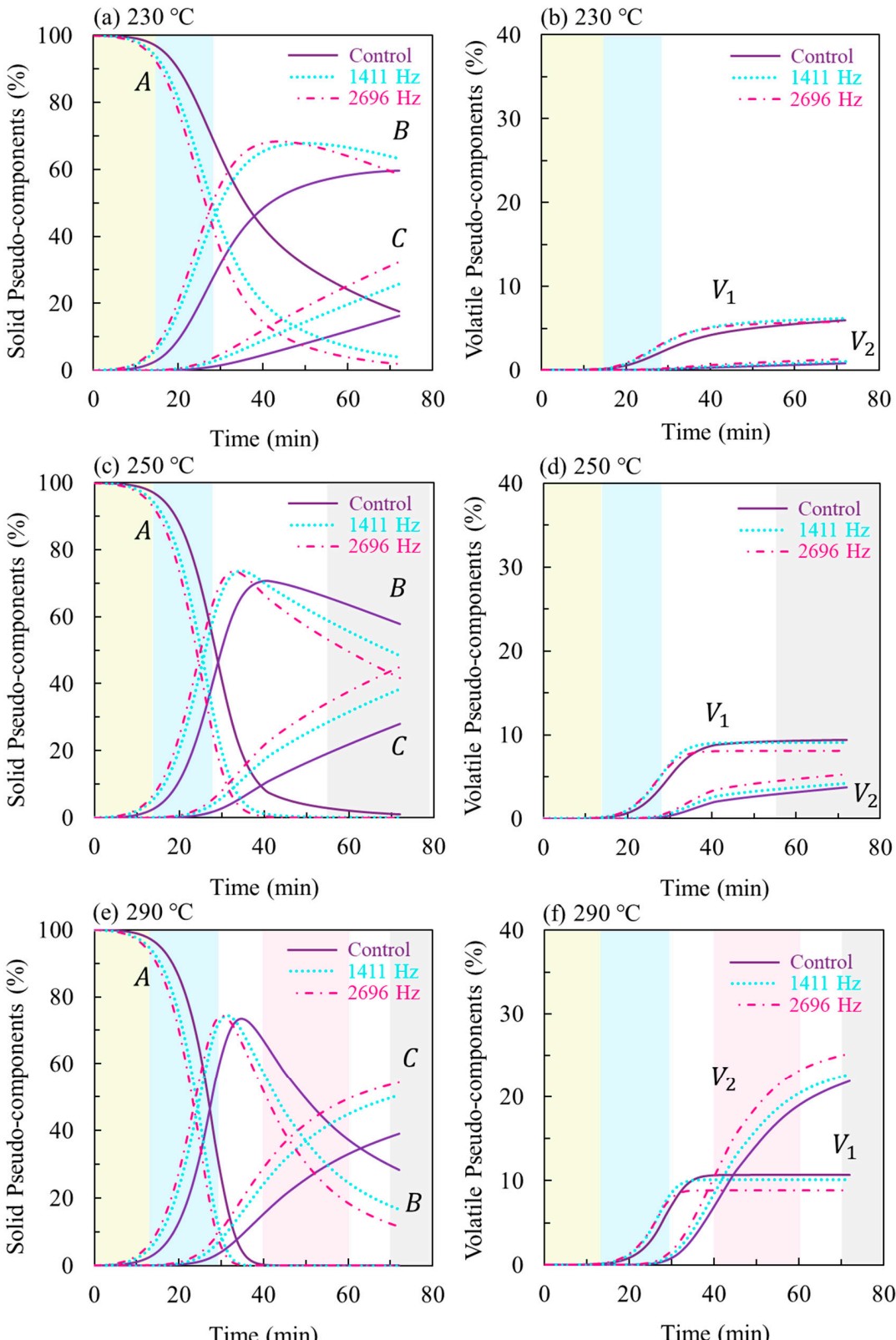

**Figure 6.** Solid and volatile pseudo-components evolution for control and acoustic 1411 and 2696 Hz treatments. (**a**,**b**) 230 °C, (**c**,**d**) 250 °C and (**e**,**f**) 290 °C.

For Severe treatment (Figure 6e,f), throughout the blue region (attribute for the catalysis region during index analysis), *A* is almost entirely consumed and *B* and $V_1$ formed. This first step reaction is

mainly attributed to hemicelluloses decomposition with a minor cellulose consumption [33]. The shift in time of thermoacoustic profiles shows the role of acoustic frequencies in anticipating hemicellulose decomposition. The slower second stage is mainly due to cellulose decomposition, with little lignin [33]. The pink region is evidenced for the 2nd DTG peak (cellulose oxidative reactions), showing the boosting effect of acoustic frequencies in the oxidizing reaction by the faster and higher releasing of $C$ and $V_2$.

## 4. Conclusions

The acoustic catalytic effect on the torrefaction process and its torrefied product has been investigated. Earlier wood degradation was evidenced. It was observed a more evident effect in enhancing degradation rates at 230 °C, decreasing with increasing temperatures. The higher treatment temperatures presented a second stage of decomposition that could be related to biomass oxidation. The catalytic effect of acoustic in increasing oxidation could be due to faster particle velocities around the wood sample, promoting higher surface oxidation and intensifying the internal heat and mass transfer. An adimensional torrefaction catalytic effect was introduced, allowing the identification of the two distinct catalytic regions. The final product analysis results did not show significant differences for proximate and HHV analysis. However, the higher thermal stress of the woody anisotropic structure due to the higher temperature gradients promoted lower compression strength values for both acoustic frequencies. The acoustic coupling kinetic simulation resulted in faster conversion rates leading to a highly enhanced first step reaction (hemicellulose) and a lesser enhancement of the second (cellulose oxidation) for acoustic conditions. Concerning mild to severe torrefaction, the thermoacoustic design could reduce residence time and improve wood particle's grindability reducing processes energy consumption and adding value to the torrefied product.

From these results, the construction of a new pilot unit has started. The obtained results will guide further adjustments on torrefaction parameters, acoustic intensities, and reactor design. Experiments exploring a cylindrical reactor configuration, higher acoustic intensities (ultrasound), volatile inline analysis, and acoustic effect on the wood components' thermal behavior will be subject to further investigation in a forthcoming study.

**Author Contributions:** E.A.S.: Reactor device concept and validation, data analysis, writing, and editing draft; L.G.O.G.: Reactor device validation, data collection, and writing; L.A.d.M.: Data analysis, project administration, supervision, and writing-review of the manuscript; I.A.S.: Data collection and analysis; B.S.C.: Data collection and analysis; M.V.G.d.M.: Reactor device concept and supervision; P.R.: Reactor device concept and writing-review of the manuscript; A.C.-P.: Reactor device concept, funding acquisition, project administration, and supervision. All authors have read and agreed to the published version of the manuscript.

**Funding:** This research received no external funding.

**Acknowledgments:** This research was funded by the Brazilian National Council for Scientific and Technological Development (CNPq), Coordenação de Aperfeiçoamento de Pessoal de Nível Superior—Brasil (CAPES)—Grant number 001, Distrito Federal Research Support Foundation (FAPDF), and Fundação Universidade de Brasília FUB/UnB/DPI/DPG.

**Conflicts of Interest:** The authors declare no conflict of interest.

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
