# Peer review of "Thermo-Acoustic Catalytic Effect on Oxidizing Woody Torrefaction"

_processes, doi:10.3390/pr8111361_

Round 1

Reviewer 1 Report

This manuscript describes an original and interesting work. As such, it has the potential to be published in Processes. However, I have the following comments that the authors have to implement in the revised manuscript before publication.

1) Introduction - The connection between the aim of the work and the literature gaps has to be better described, thus giving more strength to the reason behind this work.

2) Results and discussions/Conclusions - The authors should better highlight the practical impact of the results obtained in this work.

3) Results and discussions - The model is able to give good predictions of experimental data. Can the model tell us anything more than experiments? The authors have to stress this point in the revised manuscript.

4) Conclusions - An outlook on future research work has to be given.

I'm willing to review the revised manuscript.

Author Response

Response to Reviewer 1 Comments

Point 1: Introduction - The connection between the aim of the work and the literature gaps has to be better described, thus giving more strength to the reason behind this work.

Response 1: Thank you for giving credits for the present work and for all the questions and contributions. It made the work better. The introduction of the work was modified to improve the strength of the present work. It is important to point out that it is the second work of an unprecedented study with no literature concerning the acoustic and torrefaction process together. The thermoacoustic work involving heat transfer that inspired the new technology's insight was added in the introduction section. The text was modified between lines 42-56.

New literature [6-13] was added:

  1. Turner, I.; Rousset, P.; Rémond, R.; Perré, P. An experimental and theoretical investigation of the thermal treatment of wood (Fagus sylvatica L.) in the range 200-260 °C. Int. J. Heat Mass Transf. 2010, 53, 715–725, doi:10.1016/j.ijheatmasstransfer.2009.10.020.
  2. Santos, E.A.; Martins, C.A.; Nascimento, C.L. A new approach to treating pressure oscillations in combustion instability phenomena. Appl. Acoust. 2016, 114, 27–35, doi:10.1016/j.apacoust.2016.07.006.
  3. Cintra, B.F.C. Fernandes, E.C. Thermoacoustic instabilities of lean disc flames. Fuel, 2016, 184, 973–986.
  4. Matveev, K.I.; Culick, F.E.C. A study of the transition to instability in a Rijke tube with axial temperature gradient. J. Sound Vib. 2003, 264, 689–706.
  5. Garrett, S.L. TA-1: Thermoacoustic engines and refrigerators. Am. J. Phys. 2004, 72, 11–17.
  6. Guédra, M.; Bannwart, F..; Penelet, G.; Lotton, P. Parameter estimation for the characterization of thermoacoustic stacks and regenerators. Appl. Therm. Eng. 2015, 80, 229–237.
  7. Komarov, S.; Hirasawa, M. Enhancement of gas phase heat transfer by acoustic field application. Ultrasonics 2003, 41, 289–293, doi:10.1016/S0041-624X(02)00454-7.
  8. Bennett, G.J.; Mahon, J.; Murray, D.; Persoons, T.; Davis, I. Heat Transfer Enhancement in Ducts Due to Acoustic Excitation. 7th World Conf. Exp. Heat Transf. Fluid Mech. Thermodyn. 2009.

Point 2: Results and discussions/Conclusions - The authors should better highlight the practical impact of the results obtained in this work.

Response 2: Some insight into how this new technology improved the processes, for instance, was added in section 3.1.1, 3.2 and conclusions:

  • Section 3.1.1, (lines 228-232):

“Even if further studies are still needed to determine the real potential applicability of acoustic frequencies coupled to torrefaction, this approach holds the promise of conducting thermal treatment within shorter durations or lower temperatures, especially under oxidative atmospheres, reducing, therefore, the energy consumption in torrefaction plants.”

  • Section 3.2 (lines 295-297):

"The thermoacoustic torrefaction conferred an improved torrefied product grindability. This leads to more efficient processes enhancing added final value [40]".

  • Conclusions (lines 374-377):

"Concerning mild to severe torrefaction, the thermoacoustic design could reduce residence time and improve wood particle's grindability reducing processes energy consumption and adding value to the torrefied product".

Point 3: Results and discussions - The model is able to give good predictions of experimental data. Can the model tell us anything more than experiments? The authors have to stress this point in the revised manuscript.

Response 3: Thank you for the suggestion. The numerical modelling is able to provide qualitative and quantitative information about the evaluated biomass thermal degradation mechanisms as reported in [1]. If an extensive chemical analysis is provided, the torrefied product properties for the intermediated temperatures between collected data can be obtained, as reported in [2,3]. The predicted diagrams could reduce experimental procedures and provide useful perceptions and information for the bioenergy industry.

A paragraph was added in Section 3.3 lines 310-313:

“The predicted curves could reduce experimental procedures and provide valuable insights and information concerning treatment residence time, conversion rates and biomass thermal degradation mechanisms for torrefaction industry”.

1 .        Silveira, E.A.; Lin, B.J.; Colin, B.; Chaouch, M.; Pétrissans, A.; Rousset, P.; Chen, W.H.; Pétrissans, M. Heat treatment kinetics using three-stage approach for sustainable wood material production. Ind. Crops Prod. 2018, 124, 563–571, doi:10.1016/j.indcrop.2018.07.045.

  1. Lin, B.-J.; Silveira, E.A.; Colin, B.; Chen, W.-H.; Lin, Y.-Y.; Leconte, F.; Pétrissans, A.; Rousset, P.; Pétrissans, M. Modeling and prediction of devolatilization and elemental composition of wood during mild pyrolysis in a pilot-scale reactor. Ind. Crops Prod. 2019, 131, 357–370, doi:10.1016/j.indcrop.2019.01.065.
  2. Lin, B.J.; Silveira, E.A.; Colin, B.; Chen, W.H.; Pétrissans, A.; Rousset, P.; Pétrissans, M. Prediction of higher heating values (HHVs) and energy yield during torrefaction via kinetics. In Proceedings of the 10th International Conference on Applied Energy (ICAE2018); 2019; Vol. 158, pp. 111–116.

Point 4: Conclusions - An outlook on future research work has to be given.

Response 4: Thank you for the suggestion. The text was modified in the conclusions between lines 378-383:

“From these results, the construction of a new pilot unit has started. The obtained results will guide further adjustments on torrefaction parameters, acoustic intensities, and reactor design. Experiments exploring a cylindrical reactor configuration, higher acoustic intensities (ultrasound), volatile inline analysis, and acoustic effect on the wood components thermal behavior will be subject to further investigation in a forthcoming study”.

Reviewer 2 Report

The article is extremely interesting and can be a turning point for torrefaction technology and its wide use, because can help to solve several known problems that are identified in industrial torrefaction plants.

However, and my main concern is this, I don't find in Introduction how this technology can help solve real daily problems in biomass torrefaction technologies, and how this can be operated in real life production plants.

Numbering in section 2 must be checked and corrected.

Materials and methods section must be expanded with the experimental installation description and how the tests were conducted. This is a major point of the article.

It must be explained as well, probably in discussion, how the frequencies afect the behaviour of the compounds. This is another major point of the article.

Author Response

Response to Reviewer 2 Comments

Please see the attachment for the Revised Manuscript 

Point 1: However, and my main concern is this, I don't find in introduction how this technology can help solve real daily problems in biomass torrefaction technologies, and how this can be operated in real life production plants.

Response 1: Thank you for giving credits for the present work and all the questions and contributions. It made the work better. The introduction of the work was modified to improve the strength of the present work. It is important to point out that it is the second work of an unprecedented study with no literature concerning acoustic and torrefaction process together.  It was added in the introduction the thermoacoustic work involving heat transfer that inspired the insight for the new technology. The text was modified between lines 42-56. New literature [6-13] was added:

  1. Turner, I.; Rousset, P.; Rémond, R.; Perré, P. An experimental and theoretical investigation of the thermal treatment of wood (Fagus sylvatica L.) in the range 200-260 °C. Int. J. Heat Mass Transf. 2010, 53, 715–725, doi:10.1016/j.ijheatmasstransfer.2009.10.020.
  2. Santos, E.A.; Martins, C.A.; Nascimento, C.L. A new approach to treating pressure oscillations in combustion instability phenomena. Appl. Acoust. 2016, 114, 27–35, doi:10.1016/j.apacoust.2016.07.006.
  3. Cintra, B.F.C. Fernandes, E.C. Thermoacoustic instabilities of lean disc flames. Fuel 2016, 184, 973–986.
  4. Matveev, K.I.; Culick, F.E.C. A study of the transition to instability in a Rijke tube with axial temperature gradient. J. Sound Vib. 2003, 264, 689–706.
  5. Garrett, S.L. TA-1: Thermoacoustic engines and refrigerators. Am. J. Phys. 2004, 72, 11–17.
  6. Guédra, M.; Bannwart, F..; Penelet, G.; Lotton, P. Parameter estimation for the characterization of thermoacoustic stacks and regenerators. Appl. Therm. Eng. 2015, 80, 229–237.
  7. Komarov, S.; Hirasawa, M. Enhancement of gas phase heat transfer by acoustic field application. Ultrasonics 2003, 41, 289–293, doi:10.1016/S0041-624X(02)00454-7.
  8. Bennett, G.J.; Mahon, J.; Murray, D.; Persoons, T.; Davis, I. Heat Transfer Enhancement in Ducts Due to Acoustic Excitation. 7th World Conf. Exp. Heat Transf. Fluid Mech. Thermodyn. 2009.

Some insight into how this new technology improved the processes, for instance, was added in section 3.1.1, 3.2 and conclusions:

  • Section 3.1.1, (lines 228-232):

“Even if further studies are still needed to determine the real potential applicability of acoustic frequencies coupled to torrefaction, this approach holds the promise of conducting thermal treatment within shorter durations or lower temperatures, especially under oxidative atmospheres, reducing, therefore, the energy consumption in torrefaction plants”.

  • Section 3.2, lines 295-297.

"The thermoacoustic torrefaction conferred an improved torrefied product grindability. This leads to more efficient processes enhancing added final value [40]".

  • Conclusions, (lines 374-383):

" Concerning mild to severe torrefaction, the thermoacoustic design could reduce residence time and improve wood particle's grindability reducing processes energy consumption and adding value to the torrefied product.

From these results, the construction of a new pilot unit has started. The obtained results will guide further adjustments on torrefaction parameters, acoustic intensities, and reactor design. Experiments exploring a cylindrical reactor configuration, higher acoustic intensities (ultrasound), volatile inline analysis, and acoustic effect on the wood components thermal behavior will be subject to further investigation in a forthcoming study".

Point 2: Numbering in section 2 must be checked and corrected.

Response 2: Thank you for your contribution. The numbering in section two was revised and modifications were made at lines 153 and 164.

Point 3: Materials and methods section must be expanded with the experimental installation description and how the tests were conducted. This is a major point of the article.

Response 3: Thank you for the suggestion. The experimental installation was described in our previous work [1]:

  1. Silveira, E.A.; Morais, M.V.G. de; Rousset, P.; Caldeira-Pires, A.; Pétrissans, A.; Galvão, L.G.O. Coupling of an acoustic emissions system to a laboratory torrefaction reactor. J. Anal. Appl. Pyrolysis 2017, 129, 29–36, doi:10.1016/j.jaap.2017.12.008.

  For a better understanding of the experimental set up in this paper, a new Figure 01 (reactor set up) was added. The concerning information and description of Figure 01 are given between lines 102-122. The acoustic characterization of the system was already published in [1]. The author thinks that adding the acoustic characterization to the present work will add too much information into a single paper.

Due to the inclusion of a new figure, all the figures had to be renumbered. Modifications were made in lines:

184, 194, 213, 240, 249, 252, 257, 261, 263, 268, 279, 284, 287, 290, 300, 302, 315, 322, 325, 328, 343, 350.

Point 4: It must be explained as well, probably in discussion, how the frequencies afect the behaviour of the compounds. This is another major point of the article.

Response 4:

In the published work [1] the reactor acoustic characterization process was detailed, and the reason for the definition of frequencies was explained. Different frequencies produce different acoustic fields within the reactor, modifying the behaviour (particle velocities) at the location where the sample is positioned. Each frequency has associated a vibration mode of the cavity and a pressure field relative to that mode. Different frequencies produce different pressure field patterns. In conclusion, it was pointed that: "The catalytic effect of acoustic in increasing oxidation could be due to faster particle velocities around the wood sample, promoting higher surface oxidation and intensifying the internal heat and mass transfer." The effect on the behaviour of the wood components will be subject of a forthcoming study, analyzing the wood particle and its separate wood components with an inline analysis of volatile releasing. This will provide the kinetics of each wood components, as well as the release of different organic components for different treatment conditions (temperature and frequencies). The information about the forthcoming study was added to the conclusions between lines 378-383.

Round 2

Reviewer 2 Report

Excellent work. Parabéns.

I suggest the article to be published as it is now. The authors answered all my questions in a very assertive way.

Thank you very much. I really enjoyed reading this article.